# Subyearling Chinook salmon diets in Lower Columbia River estuarine habitats

Kerry Accola[1]*, Jeff Cordell[1], Bob Oxborrow[1], Jason D. Toft[1], Alyssa Suzumura[1], Jeffrey Grote[2]

1 School of Aquatic and Fishery Sciences, University of Washington, Seattle, Washington, United States of America, 2 Astor Environmental, Astoria, Oregon, United States of America

* kaccola@uw.edu

## Abstract

The Lower Columbia River and Estuary is critical rearing habitat for juvenile Pacific salmon. Extending from the river mouth to Bonneville Dam 235 river kilometers upstream, the Estuary has been altered by dams, dikes, and habitat loss due to deforestation and wetland removal. Since 2008, five sites have been monitored to identify the long-term status and trends in Lower Columbia River and Estuary juvenile salmon rearing habitat. Here, we address predominantly natural origin juvenile Chinook *Oncorhynchus tshawytscha* diets and identify spatial and temporal trends in prey consumption, stomach fullness, energy consumption, and the metabolic costs associated with fish size and water temperatures. Juvenile Chinook diets consisted mainly of corophiid amphipods, chironomid dipterans, and cladocerans, with other insects filling in most of the remainder of their diets. Juvenile Chinook salmon diets were stable over time, and stomach fullness and caloric intake was comparable among the sites where most salmon were collected. Juvenile Chinook salmon were frequently in water temperatures above fitness thresholds and higher water temperatures and metabolic rates will require increased foraging as water temperatures rise. Reduced growth, earlier migration, and mismatches between prey production and foraging are near term possibilities. Juvenile salmon rearing resiliency in the estuary can be aided by maintaining sufficient river discharge levels for salmon, and by restoring habitat and habitat connectivity to the mainstem channel.

## Introduction

Estuaries provide crucial nursery functions for young salmon, including refuge and growth opportunities, as they undergo osmoregulatory transitions from fresh to saline waters [1–4]. Growth during their first year of life is linked to marine survival and spawning [5,6] and the highest proportional growth for salmonids, as visual predators [7], occurs during the juvenile life stage [8]. Larger fish can better evade predation, are less susceptible to starvation, and are more efficient foragers [9]. Juvenile salmon

**Data availability statement:** All "Subyearling Chinook salmon diets in Lower Columbia River estuarine habitats" files are available from the Dryad database and are publicly accessible via this link: https://doi.org/10.5061/dryad.c59zw3rm2 Data used in this study is also visualized at https://public.tableau.com/app/profile/ecosystem.monitoring.program/viz/MacroinvertebrateCommunities/WelcometotheMacroinvertebrateEcosystemMonitoringDashboard.

**Funding:** This study was funded under Contract #11-2024 by the Northwest Power and Conservation Council/Bonneville Power Administration, to support data collected by the EMP (implemented by LCEP), and to inform regional habitat restoration efforts and action effectiveness monitoring. The funders had no role in study design, data collection and analysis, decision to publish, or preparation of the manuscript.

**Competing interests:** The authors have declared that no competing interests exist.

estuarine residency is varied, but a few species, like Chinook Salmon *Oncorhynchus tshawytscha*, are estuary-dependent [1], with a broad range of residence times [10]. Varying life history types [11,12] and subyearling migration patterns in salmon populations [13] aid adaptation to environmental variability [14]. However, habitat modifications alter prey availability, habitat connectivity, refuge, residency [3,15], and impact density-dependent foraging efficiency [16], so restoring estuarine habitats in conjunction with harvest and hatchery management can aid resiliency in the face of changing climate regimes [15].

The Columbia River (Oregon and Washington, USA) has historically been one of the highest global producers of Pacific Salmon *Oncorhynchus* spp. [17]. Consequentially, the Lower Columbia River and Estuary (LCRE) is recognized as important juvenile salmon rearing habitat [18–21]. Smaller sizes of subyearling Chinook salmon are the most prevalent species in shallow water habitats in the lower estuary [19,20,22]. Regionally, Chinook salmon are an economically, culturally, and ecologically important species, and the Columbia River is a key source of the species. Historically, spring flooding transported nutrients (wetland-derived prey), cold water, and sediments downstream to nourish migrating juvenile salmon [23], aiding foraging opportunities for their required growth enroute to the ocean [3,8]. However, anthropogenic changes beginning in the 1860s including agriculture, dike building, and dam construction for the hydropower system, altered water quality and habitat conditions [23–25], resulting in loss of 68–70% of vegetated tidal wetlands [26]. Habitat loss undermines the estuary's ability to support healthy juvenile salmon populations, and the need to protect and recover habitat for threatened and endangered evolutionarily significant units of salmon was recognized [27,28]. In 1995, the LCRE, extending from the river mouth to Bonneville Dam (235 river kilometers (RKm)), was designated an "Estuary of National Significance" [23].

Juvenile salmon need to grow as much as possible before entering the ocean in all estuaries [15], but in the LCRE, where protective marine embayments are lacking compared to places like Puget Sound or the Strait of Georgia, this growth is especially important [22]. Although loss of shallow water habitat has reduced macrodetrital contributions in the LCRE [29], food webs supporting juvenile salmon in the LCRE are still primarily detritus based [30]. Tidal wetlands are a primary source of salmon prey in the wetlands and in the estuary, as salmon disproportionately favor these allochthonous prey sources [30]. The most important prey in LCRE subyearling Chinook diets have been adult dipterans (mostly Chironomidae; non-biting midges) and amphipods (*Americorophium salmonis* and the more saline-tolerant *A. spinicorne*) [31,32], which are generally more prevalent in diets at tidally inundated sites [33–35]. Cladocerans (water fleas) have become more prevalent in mid-estuary diets [35]. The large biomass of chironomid larvae make them energetically important [36], and although cladocerans are less nutritious, they are prolific, large bodied, and slow moving, making them more accessible than other prey [37,38].

Shortly after the LCRE was deemed an estuary of national significance, the Lower Columbia Estuary Partnership (LCEP) was established. The LCEP works to protect and restore the LCRE, including by managing the Ecosystem Monitoring Program

(EMP), a multi-agency status and trends program that studies juvenile salmon rearing conditions at five sites in the LCRE [23]. Since 2008, EMP scientists have collected and analyzed key biotic and abiotic data related to ecological conditions at tidally influenced estuarine and freshwater emergent habitats used by migrating juvenile Chinook salmon. Data collection includes primary production, zooplankton, fish diets, prey resources, habitat and hydrology, and other abiotic factors. EMP estuarine sites are minimally disturbed and are a subset of the eight hydrogeomorphic reaches across the LCRE [23,39,40]. The sites are currently located at Ilwaco Slough (RKm 6), Welch Island (RKm 53), Whites Island (RKm 72), Campbell Slough (RKm 149), and Franz Lake (RKm 221) (Fig 1).

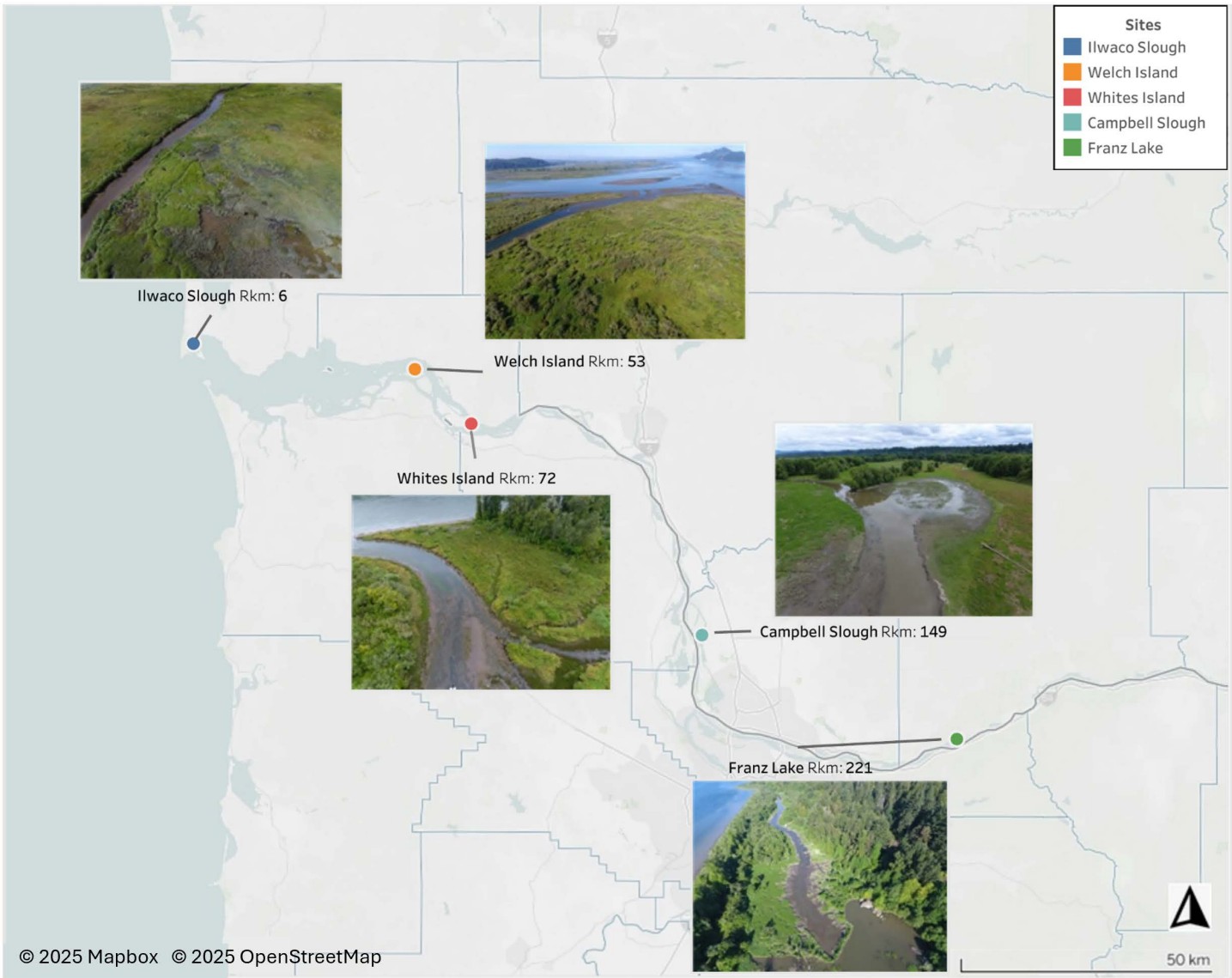

**Fig 1. Map of Lower Columbia Estuary Partnership Environmental Monitoring Program sites.** Map includes site photos, showing river kilometer distance (Rkm) from mouth of the estuary. This Tableau Public map was created by the Wetland Ecosystem Team at the University of Washington, in conjunction with the Lower Columbia Estuary Partnership, in 2025. Basemap © Mapbox, © OpenStreetMap. https://public.tableau.com/app/profile/ecosystem.monitoring.program/viz/MacroinvertebrateCommunities/WelcometotheMacroinvertebrateEcosystemMonitoringDashboard.

Here, we focus on subyearling Chinook salmon diets (78% wild, 22% hatchery) at the five EMP sites to better understand growth in the estuary [41,42]. Diet data has been collected specifically for juvenile Chinook salmon because of their Endangered Species Act listing [27], estuary dependence, and prevalence in the lower estuary [6], manifested by the various life history patterns present throughout the year [4,21,32]. We identified juvenile Chinook foraging trends, including top prey, among sites and through time. We quantified the influence of site and month on salmon fitness via their stomach fullness, energy consumption, and metabolic costs, by assessing their diets and the energetic benefits the salmon derive from them. By identifying trends in this data, we can better understand the role of estuary prey in Chinook salmon fitness.

## Methods

### Sites

Juvenile Chinook salmon diets and temperature data were analyzed from five EMP sites in the LCRE. Ilwaco Slough (RKm 6) is marsh habitat near the mouth of the Columbia River with low vegetative species richness [35]. It is the most tidally influenced site, with annual average salinities of <10 practical salinity units [23] (Fig 1). Most salmon caught here are chum salmon *O. keta* [43]. Upstream, Welch Island (RKm 53) is tidally influenced, high marsh habitat with small tidal channels. Whites Island (RKm 72), created from dredge material, is also high marsh habitat influenced by tides and prolonged freshets. Welch and Whites Islands have high vegetative species richness, and most sampled fish are juvenile Chinook salmon [35]. Campbell Slough (RKm 149) monitoring occurs at an emergent marsh that is 1.5 km from the main-stem channel, is frequently grazed by cattle, and has low vegetative species richness. The site is influenced by Bonneville Dam discharge, but pH levels frequently exceed water quality benchmarks [44], and water temperatures regularly exceed salmon fitness standards [35]. The most common salmon caught at Campbell Slough is juvenile Chinook salmon. The Franz Lake (RKm 221) sampling location, located 350 m from the mouth of the channel, is high marsh habitat impacted by beaver dams. Water levels are controlled by Bonneville Dam discharge and water quality is considered to be suitable [35]. Juvenile Chinook and coho *O. kisutch* salmon are the most frequently caught salmonids. Campbell Slough and Franz Lake have the highest fish diversity, buoyed by warm-water tolerant introduced species, some of which have life stages that prey on juvenile salmon [35,40].

### Field and laboratory

Since 2008, subyearling Chinook salmon have been collected for diet and other analyses using boat or raft-deployed 38 x 3-m variable mesh bag seines (10.0 mm and 6.3 mm wings, 4.8 mm bag). The fish were collected under Washington State Collection Permits RCW 77.15.660 and WAC 220-200-04, and under NMFS Northwest Fisheries Science Center Scientific Research Permit 22944. Sampled fish were maintained in buckets with oxygenated, in situ water, with ice added, as necessary, to maintain water temperatures at levels conducive to fish comfort. Trained biologists quickly euthanized fish with a lethal dose of MS-222, placed them on ice to slow digestion for additional analyses not described here. The stomachs were preserved in 10% formalin. In the laboratory, entire stomachs were analyzed, and each prey taxon was counted, blotted dry, weighed to the nearest 0.0001 g, and identified to the finest possible taxonomic unit (crustaceans to genus or species and insects to family or order). Water quality has been continuously measured at EMP sites using Yellow Springs Instruments (YSI) equipped with water temperature, specific conductance, pH, and dissolved oxygen probes.

### Analyses

Important components of juvenile Chinook diet composition were determined by calculating an Index of Relative Importance (IRI), which accounts for the numeric composition (counts) of each prey taxa, the gravimetric composition (weight),

and the frequency of occurrence (F; ratio of an individual prey taxa occurrence and sample size) [45]. Together, the prey abundance, weight, and frequency of occurrence in the diet were calculated as:

$$Index\ of\ Relative\ Importance = F * (\%\ Numeric\ Composition + \%\ Gravimetric\ Composition)$$

Using IRI results, we used nonmetric multidimensional scaling (NMDS) on a Bray-Curtis dissimilarity matrix (appropriate for percentage data) to visualize dissimilarities of top diet taxa among sites. Our ordination observations led to analysis of similarities to explain statistical differences between groups, and we calculated the cumulative contribution of each species, listing the taxa that account for more than 70% of the differences between sites, i.e., those taxa that were most important in driving differences in the diets among sites.

We visually categorized, then calculated, fish foraging performance, or stomach fullness, by instantaneous ration (IR), which was calculated as the ratio of total prey weight (sum of weights of all individual taxa in the diet) to total fish mass. To understand how prey in diets nutritionally benefit salmon, we quantified energy ration (ER), that is, energy consumption, in which individual prey taxon masses for each fish were multiplied by the energy density (kJ g$^{-1}$ wet mass) of each prey taxon, summed, and divided by the individual fish mass. Energy counts of prey taxa were compiled and acquired from [46] and [16]. Metabolic costs for fish were measured as maintenance metabolism ($J_m$) [47], in which $j_m$ is the mass specific maintenance costs at 0ºC (0.003), $d$ is the temperature coefficient for biomass assimilation (0.68), $t$ is water temperature in ºC, and $W$ is fish mass.

$$IR = \frac{sum\ of\ prey\ weight}{fish\ weight} \qquad ER = \frac{sum\ of\ prey\ energy\ density}{fish\ weight} \qquad J_m = j_m * e^{dt} * W$$

Finally, we used generalized linear mixed models to compare spatial and temporal IR, ER, and maintenance metabolism values to understand spatiotemporal cost-benefits for juvenile Chinook. We used gamma distributed (log link) mixed models for the zero-bounded, continuous, skewed data. The models were parameterized in terms of mean $\mu$ and a gamma dispersion parameter $\alpha$ with the structure:

$$\frac{1}{\left(\frac{\mu}{\alpha}\right)\alpha^2 * \Gamma(\alpha)} * x^{(\alpha-1)} * e^{-(x/(\frac{\mu}{\alpha}))}$$

We performed diagnostic tests on data and residuals and tested interactive and random effects. Potential variables included sites, time (month and year), fork length bin (30–65 mm and 66–99 mm; several 100 + mm salmon were excluded from analyses), mark status to distinguish marked hatchery fish from wild or unmarked fish (marked or unmarked; indicated by presence/absence of an adipose fin) the best estimate of fish origin, and water temperature at time of fish sampling.

Our best models computed IR, ER, and maintenance metabolism values among sites and months, and by fish length bin (for IR, ER models) and mark status (for maintenance metabolism models), while accounting for temporal variation that occurs among and within years, that is, the random effects of months nested within years. Mark status, rather than fork length, was used for maintenance metabolism models, because maintenance metabolism is a function of fork length. Because sample sizes were unequally dispersed among mark/unmarked salmon and fish fork lengths (most smaller study fish were wild fry, most hatchery fish were larger), interactive effects of the two would be biased towards smaller and unmarked fish. Rank deficiencies, as each site was not sampled every month, rendered the interactive effects among sites and months nonviable.

Our statistical analyses were calculated using R Statistical Software [48; version 4.2.2.] and R Studio [49]. Final models were developed using the package 'glmmTMB' [50; version 1.1.7] and were chosen considering corrected Akaike

information criterion and by considering the ecological questions we wanted to answer. Multivariate analyses on prey taxa were conducted using the package 'vegan' [51; version 2.6.4]. We organized data using "dplyr" [52; version 1.1.2]. Post-hoc analyses of mixed models included pairwise comparisons of estimated marginal means and calculating the effect size of the comparison, using "emmeans" [53; version 1.8.8]. We used "ggplot2" [54; version 3.4.4] for data visualization. Our study protocols were reviewed and granted written approval through our federal permit (NOAA Section 10 #22944-2R), which serves as the ethics board following Institutional Animal Care and Use Committee guidelines.

## Results

Water temperatures averages ranged from 4.4ºC – 10.9ºC in February to 18.9ºC – 23.3ºC in July. In June, the water temperature average ranged from 14.9ºC – 20.8ºC. The maximum temperature averages at Franz Lake and Ilwaco Slough were 14.8 ºC. Most high temperatures occurred at Welch Island, Whites Island, and Campbell Slough, where the majority of juvenile Chinook were sampled (Fig 2).

### Spatial and temporal distribution

A total of 1365 juvenile Chinook salmon diets were analyzed from 2008–2021 EMP fish samples. 18 fish with empty stomachs were removed from analyses. Fish were selected for diet analyses to maximize representation from sites, months, and fish lengths. Most Chinook salmon (91%) used for diet analyses were netted at the middle three EMP sites (Campbell Slough ($n=343$), Welch Island ($n=444$), and Whites Island ($n=455$), with small overall sample sizes at Franz Lake ($n=83$) and Ilwaco Slough ($n=40$). 78% ($n=1065$) of sampled juvenile Chinook were unmarked and considered to be natural origin (wild) fish. Juvenile Chinook counts were highest in May ($n=593$), and most salmon were in the smallest length category (30–65 mm fork length; $n=775$) (Table 1; see also [20]). In general, most smaller study Chinook were wild fry, most hatchery Chinook were larger fingerlings due to release protocols, yet larger fish were a comparable mixture of wild and hatchery origin.

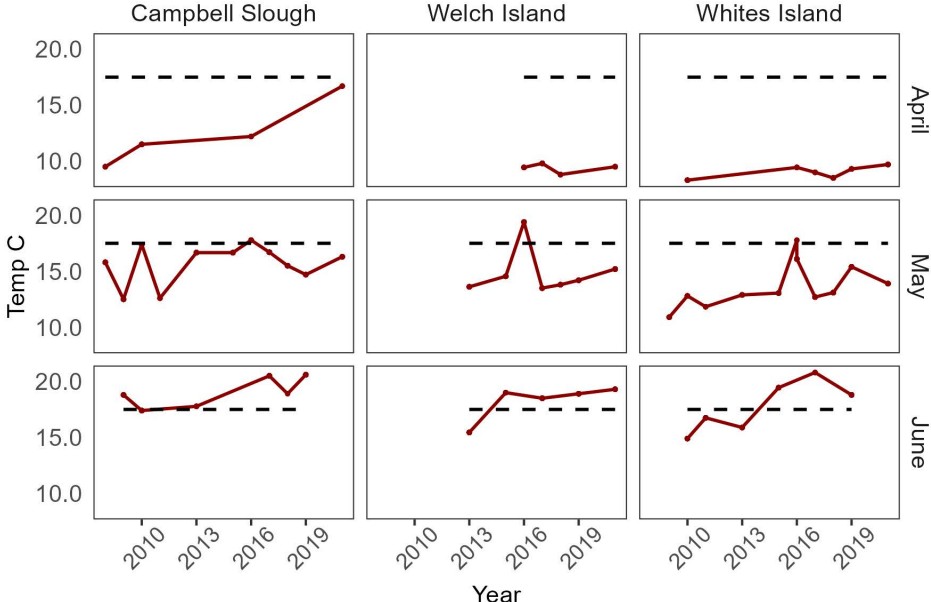

**Fig 2. Line plots of water temperature measurements (ºC) among top EMP sites and months.** Top sites and months had the highest juvenile Chinook densities. Black dotted line represents 17.5ºC, which [44] recommends as the limit (seven day average of maximum water temperatures) for Chinook rearing.

**Table 1.** Counts of Juvenile Chinook salmon selected for diet analyses.

| | Ilwaco Slough | Welch Island | Whites Island | Campbell Slough | Franz Lake |
|---|---|---|---|---|---|
| February | 29 | 25 | 5 | 4 | 5 |
| March | 5 | 62 | 44 | 8 | – |
| April | 6 | 79 | 85 | 27 | 36 |
| May | – | 154 | 186 | 216 | 42 |
| June | – | 98 | 108 | 75 | – |
| July | – | 26 | 27 | 13 | – |
| Length (mm): 30–65; 66–99 | 40; 0 | 308; 136 | 285; 170 | 91; 252 | 59; 24 |
| Unmarked; Marked | 40; 0 | 383; 61 | 400; 55 | 180; 163 | 62; 21 |
| **Total** | **40** | **444** | **455** | **343** | **83** |

Salmon are sorted by sites, month, length (30–65 mm, 66–99 mm), and mark status (marked=hatchery, unmarked=wild). Counts include salmon selected for diet analyses and do not reflect total fish sampled.

## Diet composition

Prey taxa with the highest IRI in juvenile Chinook diets, for all fish fork lengths (30 mm – 99 mm), were amphipods, cladocerans, and dipterans (Fig 3). Juvenile Chinook primarily consumed chironomids and other dipterans and insects at the most upstream site (Franz Lake), included cladocerans at Welch Island, Whites Island, and Campbell Slough (although predominantly at Campbell Slough), and consumed a mixture of mostly corophiid amphipods and chironomid dipterans at Whites Island, Welch Island, and Ilwaco Slough [35; see also [31]. Ilwaco Slough diets were a mixture of amphipods (*Americorophium* spp.) and primarily Chironomidae and Psychodidae dipterans.

*Americorophium* spp. (mostly *A. salmonis*, then *A. spinicorne*) comprised 75% of the amphipod counts and weights and were predominant in Welch and Whites Islands' salmon diets. The remaining amphipods, in order of count and weight importance, were unidentified amphipods, *Eogammarus confervicolus*, and *Hyallela azteca*. *Daphnia* spp. (water fleas) comprised 67% of the counts and weights of cladocerans. Within dipterans, chironomids accounted for 47% of the counts and 39% of the weights. Overall, chironomids and unidentified dipterans, which may be chironomids, comprised 96% of the dipterans present in our juvenile Chinook diets. Adult/emergent chironomids were the most numerous chironomid life stage (44%), and pupa made up the most weight (60%) in the diets.

Diet patterns were portrayed by an NMDS ordination using IRI values to show salmon diet differences among sites, by year, and between size length bins (Fig 4). The fourth most important diet taxon was Hemiptera, followed by the remaining taxa labelled in Fig 3. Fish netted at Whites Island, Welch Island, and Ilwaco Slough had more similar diets, and Campbell Slough and Franz Lake diets had dipterans in common, but differed based on the presence or absence of cladocerans. Analysis of similarities (ANOSIM) and pairwise similarity percentage (SIMPER) site comparisons corroborate the ordinations through statistical analyses, by demonstrating that Campbell Slough and Franz Lake diets were dissimilar from the remaining three sites by the prevalence of amphipods downstream and dipterans and cladocerans upstream (Table 2). Pairwise comparisons show no diet prey differences among different fish length bins. There were no long-term diet differences among years, except that cladocerans did not become prevalent in juvenile Chinook diets until 2016. A few years had temporal diet differences that were likely due to sampling anomalies (e.g., attenuated sampling in 2020 due to COVID-19).

## Diet benefits and costs

**Fish fullness.** Fish fullness, measured by instantaneous ration, was highest at Ilwaco Slough and lowest at Franz Lake (Fig 5). However, sample sizes were small at these sites due to sampling challenges (Franz Lake) and site location

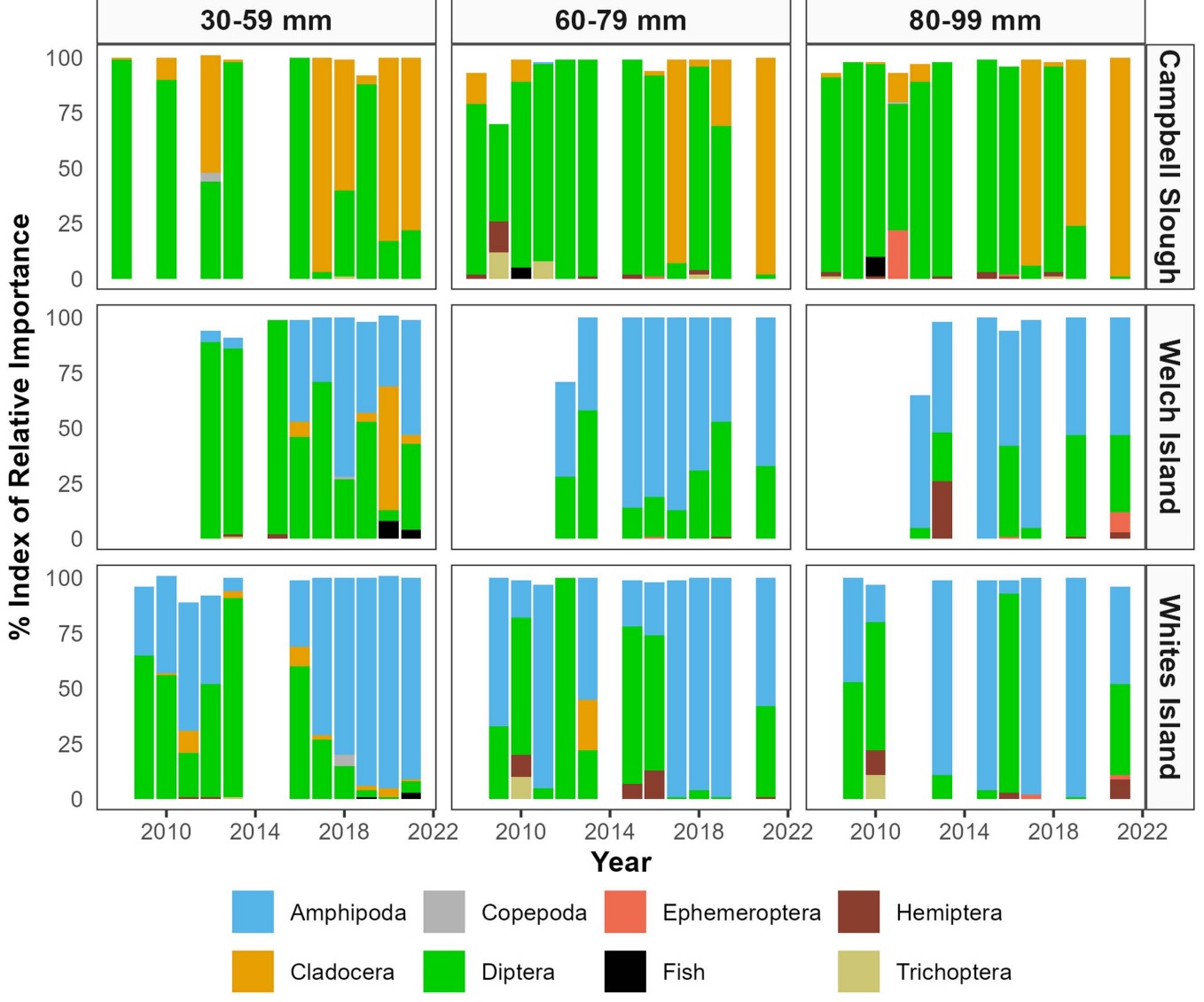

**Fig 3. Stacked bar plots representing top juvenile Chinook salmon prey.** Prey are sorted by year, fork length bin, and site. Top prey are color sorted by taxa and plot the Index of Relative Importance (IRI) values for taxa each year. IRI accounts for the counts, weights, and frequency of occurrence of prey in the diet.

(Ilwaco Slough). 2008–2021 Ilwaco Slough diets consisted entirely of unmarked fry. Stomach fullness was comparable among Welch Island, Whites Island, and Campbell Slough, and there was no statistical difference in fish fullness among months. Smaller Chinook had higher fullness marginal mean values than larger fish, so accounting for their body weight, they were filling their stomachs more than larger fish (effect size = 0.39; Table 2). Overall, study fish do not appear to be prey-limited (Fig 6).

**Caloric intake.** Ilwaco Slough Chinook stomachs, dominated by amphipods, produced the highest energy ration values. Franz Lake diets, consisting mainly of dipterans and other wetland insects, had the lowest energy ration values, likely because of lower fullness levels compared to other sites (Fig 5, Table 2). Energy intake was comparable in diets

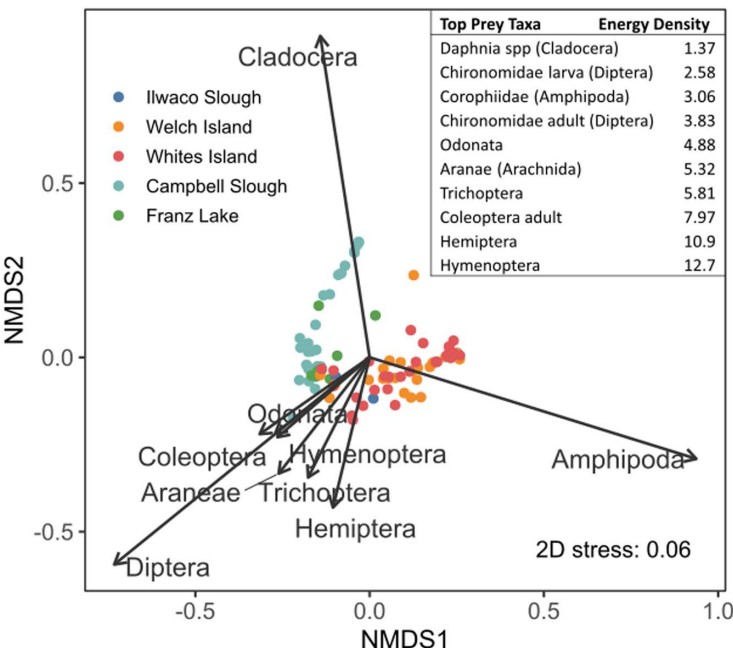

**Fig 4. Two-dimensional NMDS plot of Index of Relative Importance (IRI) for major prey.** NMDS plot shows top prey in juvenile Chinook diets between 2008 and 2021. Each point represents fish diets binned by length (30−59 mm, 60−79 mm, 80−99 mm) for each year at each site (color-coded). Black vectors depict significant species loadings (p = 0.05), with longer arrows indicating a stronger association between the species and IRI. The strongest weights are for Amphipoda, Diptera, and Cladocera, signifying they are the most important salmon prey. Inset: energy density values (kJ g⁻¹) for top prey taxa.

among Welch Island, Whites Island, and Campbell Slough. Accounting for weight, smaller fish had higher caloric intake than larger fish (effect size = 0.39), demonstrating that they derived more nutritional benefit from the prey they were consuming. July was the only month with higher energy ration marginal mean values than other months (Table 2). Higher July caloric intake was likely due to highly nutritious prey including Trichoptera, Hemiptera, and Hymenoptera replacing some amphipods in the Welch and Whites Island's diets, and fish and insects replacing cladocerans in Campbell Slough diets (see also [55]). Larger wild and hatchery fish fared similarly in diets.

**Metabolic costs.** Metabolic costs increased month to month because larger, growing fish have higher metabolic requirements and water temperatures, which increase gastric evacuation rates [56], rise as the year progresses. Higher water temperatures increase metabolic costs more than other factors [56,57] and fish must compensate by increasing their energy intake [56]. Campbell Slough had the highest metabolic costs among the study sites and regularly has water temperatures that exceed salmon fitness thresholds [35]. Most Campbell Slough salmon were larger fish, in large part due to hatchery fish prevalence, while Welch and Whites Island's salmon had greater size diversity. Marked (hatchery) fish had higher metabolic costs, but this was influenced by the marked fish category having more large fish.

## Discussion

Since 2008, we monitored trends of prey quantity, quality, and diet benefits for subyearling Chinook salmon at five minimally disturbed sites in the heavily modified Lower Columbia Estuary [58]. Overall, juvenile Chinook diets at EMP sites appear to be stable, even in current sub-optimal temperature regimes, with salmon experiencing foraging opportunities in which they balance their diets depending on prey availability.

**Table 2. Post-hoc pairwise tests on the estimated marginal means (MM) values of stomach fullness, caloric intake, and metabolic costs.**

| Sites | Stomach Fullness | | | | Caloric Intake | | | | Metabolic Costs | | | | Diet Composition | |
|---|---|---|---|---|---|---|---|---|---|---|---|---|---|---|
| | MM Comp | L 95% CI | U 95% CI | Effect Size | MM Comp | L 95% CI | U 95% CI | Effect Size | MM Comp | L 95% CI | U 95% CI | Effect Size | ANOSIM p & R value | SIMPER |
| Iwaco – Welch | 0.59 | 0.07 | 1.11 | **0.67** | 0.83 | 0.29 | 1.36 | **0.91** | 0.09 | -0.23 | 0.40 | 0.17 | | |
| Iwaco – Whites | 0.54 | 0.02 | 1.06 | **0.62** | 0.84 | 0.30 | 1.37 | **0.92** | 0.03 | -0.29 | 0.35 | 0.05 | | |
| Iwaco – Campbell | 0.51 | -0.02 | 1.04 | 0.58 | 0.73 | 0.18 | 1.29 | **0.81** | -0.40 | -0.73 | -0.08 | **-0.81** | 0.05; 0.50 | A, D |
| Iwaco – Franz | 1.05 | 0.50 | 1.61 | **1.20** | 1.15 | 0.59 | 1.72 | **1.27** | 0.32 | -0.01 | 0.65 | 0.64 | 0.46; 0.33 | A, D |
| Welch – Whites | -0.05 | -0.23 | 0.13 | -0.05 | 0.01 | -0.17 | 0.19 | 0.01 | -0.06 | -0.16 | 0.05 | -0.12 | | |
| Welch – Campbell | -0.08 | -0.29 | 0.14 | -0.09 | -0.09 | -0.32 | 0.13 | -0.10 | -0.49 | -0.61 | -0.37 | **-0.98** | 0.01; 0.47 | A, D |
| Welch – Franz | 0.47 | 0.12 | 0.81 | **0.53** | 0.33 | -0.03 | 0.68 | 0.36 | 0.24 | 0.03 | 0.45 | **0.47** | 0.01; 0.55 | A, D |
| Whites – Campbell | -0.03 | -0.23 | 0.17 | -0.04 | -0.10 | -0.31 | 0.11 | -0.11 | -0.43 | -0.54 | -0.32 | **-0.86** | 0.01; 0.50 | A, D |
| Whites – Franz | 0.51 | 0.17 | 0.85 | **0.58** | 0.32 | -0.03 | 0.66 | 0.35 | 0.29 | 0.09 | 0.50 | **0.59** | 0.01; 0.499 | A, D |
| Campbell – Franz | 0.54 | 0.20 | 0.89 | **0.62** | 0.42 | 0.06 | 0.78 | **0.46** | 0.73 | 0.52 | 0.94 | **1.45** | | |

| Lengths (mm) & Mark Status | MM Comp | L 95% CI | U 95% CI | Effect Size | MM Comp | L 95% CI | U 95% CI | Effect Size | MM Comp | L 95% CI | U 95% CI | Effect Size | ANOSIM p-value | SIMPER |
|---|---|---|---|---|---|---|---|---|---|---|---|---|---|---|
| 30-65; 66-99 | 0.34 | 0.21 | 0.48 | **0.39** | 0.35 | 0.21 | 0.49 | **0.39** | -0.62 | -0.69 | -0.54 | **-1.23** | | |
| Unmarked – Marked | | | | | | | | | | | | | | |

| Months | Stomach Fullness | | | | Caloric Intake | | | | Metabolic Costs | | | | Diet Composition | |
|---|---|---|---|---|---|---|---|---|---|---|---|---|---|---|
| | MM Comp | L 95% CI | U 95% CI | Effect Size | MM Comp | L 95% CI | U 95% CI | Effect Size | MM Comp | L 95% CI | U 95% CI | Effect Size | ANOSIM p-value | SIMPER |
| February – March | -0.11 | -0.84 | 0.62 | -0.13 | -0.04 | -0.81 | 0.73 | -0.05 | -0.32 | -0.99 | 0.35 | -0.64 | | |
| February – April | 0.05 | -0.67 | 0.78 | 0.06 | 0.06 | -0.69 | 0.80 | 0.06 | -1.02 | -1.68 | -0.35 | **-2.02** | | |
| February – May | -0.14 | -0.85 | 0.58 | -0.16 | -0.16 | -0.89 | 0.57 | -0.18 | -1.90 | -2.52 | -1.27 | **-3.78** | | |
| February – June | 0.06 | -0.69 | 0.82 | 0.07 | -0.09 | -0.87 | 0.68 | -0.10 | -2.46 | -3.10 | -1.82 | **-4.90** | | |
| February – July | -0.32 | -1.20 | 0.56 | -0.36 | -0.85 | -1.76 | 0.06 | -0.93 | -2.78 | -3.53 | -2.03 | **-5.54** | | |
| March – April | 0.17 | -0.36 | 0.70 | 0.19 | 0.10 | -0.45 | 0.65 | 0.11 | -0.69 | -1.22 | -0.17 | **-1.38** | | |
| March – May | -0.02 | -0.53 | 0.48 | -0.03 | -0.12 | -0.64 | 0.41 | -0.13 | -1.57 | -2.04 | -1.10 | **-3.13** | | |
| March – June | 0.17 | -0.37 | 0.72 | 0.20 | -0.05 | -0.61 | 0.52 | -0.05 | -2.14 | -2.62 | -1.65 | **-4.26** | | |
| March – July | -0.21 | -0.92 | 0.51 | -0.23 | -0.81 | -1.55 | -0.06 | **-0.89** | -2.46 | -3.09 | -1.84 | **-4.90** | | |
| April – May | -0.19 | -0.59 | 0.21 | -0.22 | -0.22 | -0.64 | 0.20 | -0.24 | -0.88 | -1.29 | -0.47 | **-1.75** | | |
| April – June | 0.01 | -0.45 | 0.46 | 0.01 | -0.15 | -0.62 | 0.33 | -0.16 | -1.45 | -1.92 | -0.97 | **-2.88** | | |
| April – July | -0.37 | -1.02 | 0.27 | -0.43 | -0.91 | -1.58 | -0.23 | **-0.99** | -1.77 | -2.38 | -1.16 | **-3.52** | | |
| May – June | 0.20 | -0.20 | 0.59 | 0.23 | 0.07 | -0.34 | 0.49 | 0.997 | -0.56 | -0.97 | -0.16 | **-1.12** | | |
| May – July | -0.18 | -0.78 | 0.41 | -0.21 | -0.69 | -1.32 | -0.06 | **0.022** | -0.89 | -1.45 | -0.33 | **-1.77** | | |
| June – July | -0.38 | -1.00 | 0.23 | -0.43 | -0.76 | -1.41 | -0.11 | **0.011** | -0.32 | -0.90 | 0.25 | -0.64 | | |

Tests include MM comparisons, 95% confidence intervals (CI), and effect sizes. Effect sizes that do not contain zero in their confidence intervals are highlighted in bold. Effect size (Cohen's d; medium effect = 0.5) is the pairwise difference of marginal means estimates divided by the population standard deviation. Pairwise comparisons are for Site, Month, Mark Status, and Fish Length bin. Pairwise Analysis of Similarities (ANOSIM) and similarity percentages (SIMPER) calculated on Index of Relative Importance diet data show differences among top prey. The Bonferroni p-value and R value reflect the strength of the relationship. SIMPER displays taxa that contribute at least 70% to the differences between groups. A = amphipods, D = dipterans.

## Prey trends

All size classes of sampled Chinook had similar interannual prey trends, in which three taxa (corophiid amphipods, cladocerans, and chironomid dipterans) comprised most of the diets. Rather than an even distribution among the top three prey, there was frequent interannual variation in which one taxon (typically amphipods or dipterans) dominated the year, with other top prey taxa making smaller contributions. Interannual fluctuation in top prey indicates prey availability drives diet patterns, and prey tradeoffs indicate an estuary's capacity for salmon diet adaptability [59,60]. A possible exception to prey availability driving diet patterns is that larger Chinook appeared to select for amphipods at Welch and Whites Islands even though dipterans were more prevalent than amphipods in neuston samples taken at those sites [35]. Annual fluctuations may be due to differing annual flow regimes, specifically high flow years, e.g., *Americorophium* spp. play a larger role in wetland diets when they are more readily available from high river flows scouring benthic habitats [43,61].

Chironomids and *Americorophium* amphipods are also important in main channel yearling salmonid diets, with other insects complementing their diets [62]. Wetland prey presence in main channel diets highlights the importance of tidal wetlands as prey sources to the varying Chinook life history patterns in the LCRE [20], including fast-moving main channel yearling salmon that mostly originate from interior stocks [61,62], and smaller LCEP EMP subyearling Chinook salmon that originate from Lower Columbia stocks [35].

Beginning in 2016, cladocerans became more prominent in salmon diets at Campbell Slough, with amphipods and dipterans remaining important prey throughout the study period. Cladocerans, followed by cyclopoid copepods (the latter were not important in Chinook diets), were also prominent in neuston samples at Welch Island, Whites Island, and Campbell Slough [35]. Amphipods and dipterans (predominantly chironomids) were present in lower quantities in neuston and benthic core samples [35], but their large size and nutrition likely makes them preferable to visual predators like Chinook salmon. Cladocerans may share diet prominence with dipterans at Campbell Slough because amphipods were less available, based on neuston tow and benthic core data [35]. Cladocerans are slow-moving prey that are easily captured but are less nutritious than other top prey, and they may represent as an alternative when there is a paucity of more nutritious prey. Their accessibility likely contributed to their high biomass in diets, enabling higher salmon growth rates in Campbell Slough and similar habitats, where flooding supplies detritus and nutrients, and elevated water temperatures increase phytoplankton and zooplankton productivity [31,60,63,64].

Energy-dense insects became more prevalent at the middle three LCEP EMP sites as the summer progressed, especially at Welch and Whites Islands. Although [65] found the highest energy transport from LCRE tidal wetlands was from non-chironomid dipterans and (predominantly) corixid hemipterans, unidentified dipterans (which could have been chironomids) were important in our salmon diets, but hemipterans ('true bugs') did not have as high importance in our study diets as would be expected, given their high nutritional value. The exceptions were a few outlying years, when larger fish at Welch and Whites Islands complemented their corophiid amphipod and chironomid dipteran diets with hemipterans, especially during summer months. Most hemipterans in our study were corixids, which tend to have tougher exoskeletons and be less easily digestible than other hemipterans. As aquatic invertebrates, they may be able to escape predation more easily than terrestrial insects, so it is unclear whether smaller Chinook fry have more difficulty capturing them, experience gape limitations, or it is not worth the energy expenditure to pursue and prey on them.

## Spatial and temporal trends

Most juvenile Chinook salmon (91%) were netted at the middle sites (Welch Island, Whites Island, and Campbell Slough). Challenges of sampling access at Franz Lake, which was hindered by beaver dams and water levels [35], and the location of Ilwaco Slough near the mouth of the estuary, where salmon may quickly pass through to the ocean, likely contributed to low Chinook densities at these sites. Although few juvenile Chinook were netted at Ilwaco Slough, juvenile chum counts were highest at this site and Chinook were present in low densities in spring months. Stomach fullness results from our

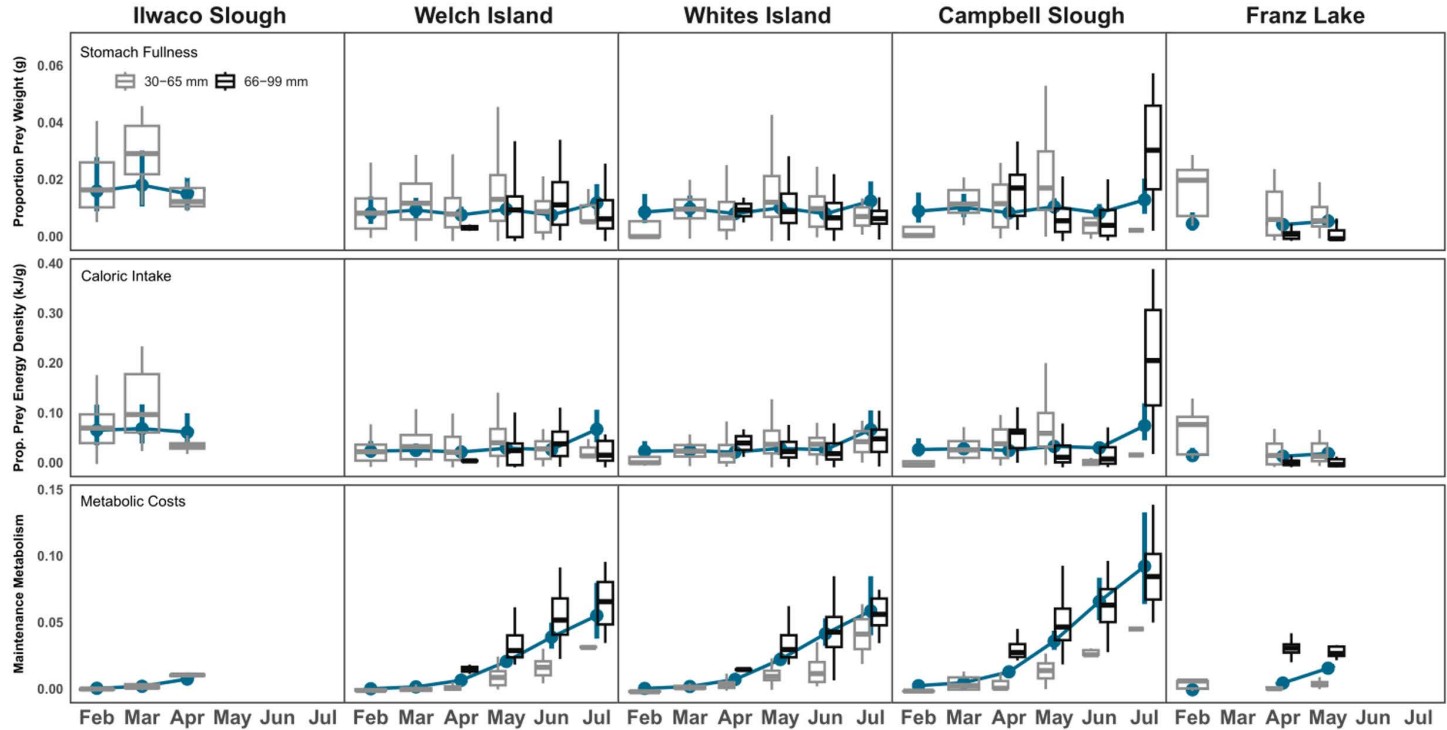

**Fig 5. Estimated marginal mean (EMM) dot plots overlaid by boxplots of raw data.** Dot plots (in blue) depict the EMMs of generalized linear mixed models' response variables (Stomach Fullness, Caloric Intake, and Metabolic Costs) plotted by Site and Month. Overlaying boxplots are raw data values that are color-sorted by fish length (30-65 mm = gray, 66-99 mm = black).

study differ from [31], who found upstream diets had the fullest stomachs, while downstream stomachs were emptier. Their downstream subyearling Chinook were offshore and migrating rapidly, while our Ilwaco Slough subyearling fish were smaller and likely utilizing off-channel habitats to increase growth before leaving the estuary.

Salmon counts peaked in May each year, then decreased throughout summer – more quickly for small fish – because some small fish transitioned to a larger size class, while others left the estuary or died. A July increase in caloric intake values indicates that salmon were acquiring nutritious, energy dense prey during this time. Although caloric intake was highest in July, stomach fullness was not. Most nutritious food was found in diets from sites where predominantly larger salmon were found (Welch Island, Whites Island, Campbell Slough). Prey included high energy dipterans, hemipterans, hymenopterans, trichopterans (all insects), and amphipods, which can help mitigate higher metabolic rates due to elevated water temperatures [66,67]. This was especially true at Campbell Slough, where reduced connectivity to the main channel in low water periods may have increased water temperatures, but also facilitated retaining wetland insects more than open sites like Welch and Whites Islands.

Juvenile Chinook salmon have exhibited steady annual presence at Welch Island, Whites Island, and Campbell Slough, even with frequent high temperatures surpassing minimal water temperature thresholds for juvenile salmon/steelhead fitness (80 days exceeding 19ºC in most years) [35]. Salmon densities decline in June, but May has continued to be the peak month of juvenile Chinook abundances sampled throughout our study period. The Washington Department of Ecology recommends a threshold of 17.5ºC as a water temperature standard [44]. [68] found that compared to warmer streams, Chinook grew faster with maximum temperatures of 16ºC. Lower water temperatures also benefit Chinook over their non-salmonid competitors [69]. However, water temperature-related related changing migration patterns for our study are not yet apparent.

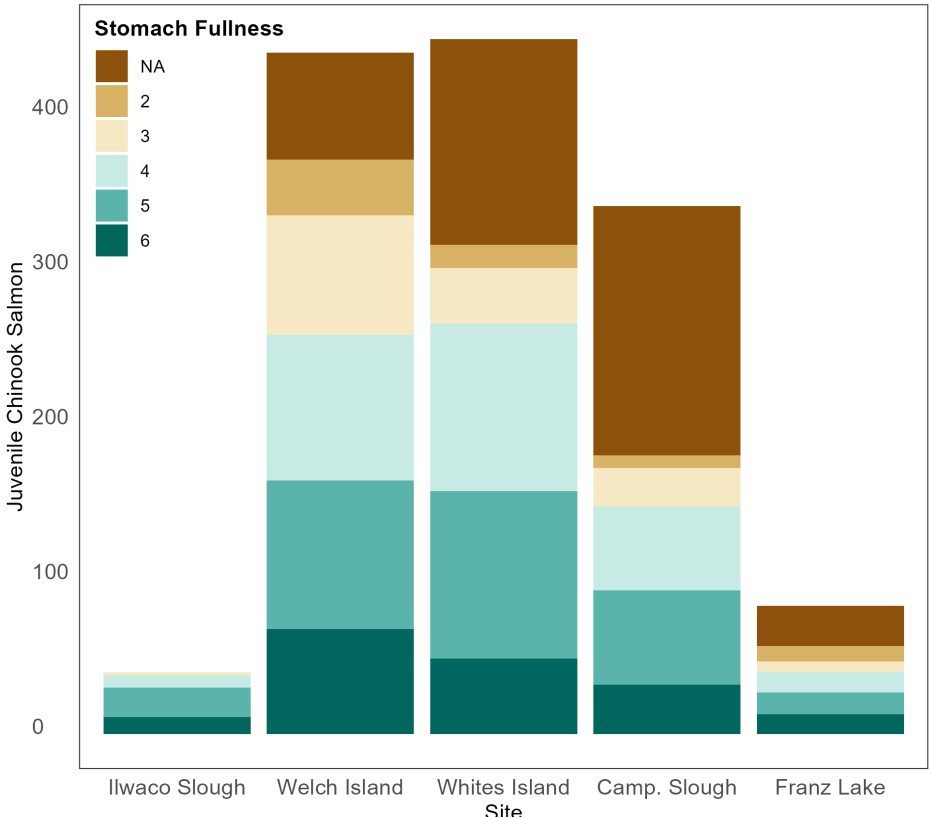

**Fig 6. Stacked bar plots portraying juvenile Chinook salmon stomach fullness.** Salmon are sorted by site and color-coded by categorization of stomach fullness. Blank means stomach fullness was not categorized. 2 indicates an emptier stomach and 6 indicates a full stomach.

The three lower estuary sites (Ilwaco Slough, Welch Island, Whites Island) are tidally driven and inundated several hours each day, while Franz Lake and Campbell Slough are off channel sites with reduced connectivity to the mainstem and are therefore more affected by freshet and dam discharge. River discharge is important in creating ephemeral flooded habitats [66,70] that moderate higher water temperatures, especially at sites with reduced connectivity to the mainstem channel. Franz Lake and Campbell Slough have higher fish diversity and richness, driven by introduced species that are tolerant of warm temperatures and low dissolved oxygen [21]. The six most common fish (five native, one non-native) that have life stages that prey on juvenile salmon are freshwater fish found primarily at Franz Lake and Campbell Slough [35].

## Summary

Agriculture, residential development, and dams have modified critical juvenile salmon rearing habitat in the LCRE [23]. As a more estuary-dependent species, juvenile Chinook are especially susceptible to estuarine modifications [1,3]. Their marine survival is positively correlated with how much the estuary is unmodified [6], because natural estuarine habits offer refuge from predation [6] and prey production for growth [15]. However, the status and trends of juvenile Chinook diets at five monitoring sites in the Lower Columbia River and Estuary shows stable diets, dominated by amphipods and dipterans, and cladocerans. Other prey, including other insects, complement the diets. Our results verify the importance of extant tidal wetlands habitats for juvenile salmon diets. One of the top salmon prey for subyearling Chinook, chironomid dipterans, are linked to both aquatic and terrestrial habitats as they progress from larvae to adults. Wetlands are sources

of other nutritious vegetation-associated prey, including other insects. [21] found juvenile Chinook use emergent marsh habitat for multiple weeks, growing 0.53 mm/day, re-entering the same habitats after vacating them during ebb tides. Re-entry indicates foraging-related movement patterns, as salmon use secondary channels against outgoing tides during peak emergence times of prey insects [21]. Main channel complexity is also important for growth, as vegetated shoreline habitats can provide insects for salmon and channel complexity can reduce flushing of salmon in high flow periods [70].

While juvenile Chinook migration currently peaks in May, increasing annual water temperatures will lead to higher energy requirements to maintain growth to compensate for increased metabolic costs, or to fish emigrating at smaller sizes than is optimal for first year survival. First year growth includes early ocean entry, so estuarine growth may be able to be postponed in system-specific situations [71], but early marine salmon have increased metabolic requirements in warm ocean regimes [67], and the LCRE lacks protective marine embayments outside the estuary. Improving estuary wide habitat conditions and cold water refugia in the LCRE will benefit juvenile salmon by easing metabolic stressors and providing high quality prey. The current state of climate change, including extreme summer heat events, highlights the need for water temperature mitigation and flooded habitats via dam discharge, channel access to the mainstem to promote and sustain healthy water temperatures, and a high-quality prey field for optimal growing conditions for juvenile salmon.

Next steps should include bioenergetics modeling and pairing diet data with stable isotopes and biophysical indicators at LCEP EMP sites to increase our understanding of estuarine influence on juvenile salmonid growth (e.g., [42,72]. Further analyses may include studying hatchery and wild Chinook diets in isolation to investigate potential diet differences and competition for key resources (see [73]), because wild salmon have higher estuarine growth [74]. Diets of pooled hatchery and wild fish with different lengths did not differ in our study, and diet contents through time were stable. Thus, though smaller fish were predominantly wild and larger fish had higher percentages of hatchery fish, we expect all juvenile Chinook in the LCRE to have similar diets.

## Acknowledgments

We thank the many partners involved in the research of the Lower Columbia Estuary Partnership (LCEP) Ecosystem Monitoring Program (EMP), including the current team of EMP research scientists that also includes Curtis Roegner, Tawnya Peterson, Joseph Needoba, and Ian Edgar. We are grateful to the LCEP oversight team, including Catherine Corbett, and formerly Sarah Kidd and Sneha Rao Manohar. We thank the team at NOAA Fisheries, including Regan A. McNatt, Susan A. Hinton, Lyndal L. Johnson, G. Curtis Roegner, Daniel Lomax, and Sean Sol, for collection of neuston and juvenile Chinook salmon samples. Finally, we thank Joe Needoba and the team at OHSU for collecting water temperature data.

## Author contributions

**Conceptualization:** Jeff Cordell.

**Data curation:** Kerry Accola, Bob Oxborrow, Alyssa Suzumura.

**Formal analysis:** Kerry Accola.

**Investigation:** Jeff Cordell, Bob Oxborrow, Alyssa Suzumura, Jeffrey Grote.

**Methodology:** Jeff Cordell, Jason D. Toft.

**Project administration:** Jeff Cordell, Jason D. Toft.

**Software:** Kerry Accola.

**Supervision:** Jeff Cordell, Jason D. Toft.

**Visualization:** Kerry Accola.

**Writing – original draft:** Kerry Accola.

**Writing – review & editing:** Kerry Accola, Jeff Cordell, Bob Oxborrow, Jason D. Toft, Jeffrey Grote.

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
