## [Decision Letter · Decision Letter 0]

PONE-D-25-06245Subyearling Chinook salmon diets in Lower Columbia River estuarine habitatsPLOS ONE

Dear Dr. Accola,

Thank you for submitting your manuscript to PLOS ONE. After careful consideration, we invite you to submit a revised version of the manuscript that addresses the points raised during the review process, which are detailed below.

We look forward to receiving your revised manuscript.

Kind regards,

Jason Magnuson, Ph.D.

Academic Editor

PLOS ONE

Journal Requirements:

2. To comply with PLOS ONE submissions requirements, in your Methods section, please provide additional information regarding the experiments involving animals and ensure you have included details on (1) methods of sacrifice, and (2) efforts to alleviate suffering.

This study was funded under Contract #11-2024 by the Northwest Power and Conservation Council/Bonneville Power Administration, to support data collected by the EMP (implemented by LCEP), and to inform regional habitat restoration efforts and action effectiveness monitoring.

6. We note that you have indicated that there are restrictions to data sharing for this study. PLOS only allows data to be available upon request if there are legal or ethical restrictions on sharing data publicly. For more information on unacceptable data access restrictions, please see http://journals.plos.org/plosone/s/data-availability#loc-unacceptable-data-access-restrictions.

7. We note that Figure 1 in your submission contain [map/satellite] images which may be copyrighted. All PLOS content is published under the Creative Commons Attribution License (CC BY 4.0), which means that the manuscript, images, and Supporting Information files will be freely available online, and any third party is permitted to access, download, copy, distribute, and use these materials in any way, even commercially, with proper attribution. For these reasons, we cannot publish previously copyrighted maps or satellite images created using proprietary data, such as Google software (Google Maps, Street View, and Earth). For more information, see our copyright guidelines: http://journals.plos.org/plosone/s/licenses-and-copyright.

Reviewers' comments:

Reviewer's Responses to Questions

**Comments to the Author**

1. Is the manuscript technically sound, and do the data support the conclusions?

Reviewer #1: Yes

Reviewer #2: Partly

2. Has the statistical analysis been performed appropriately and rigorously? 

Reviewer #1: Yes

Reviewer #2: Yes

3. Have the authors made all data underlying the findings in their manuscript fully available?

Reviewer #1: Yes

Reviewer #2: Yes

4. Is the manuscript presented in an intelligible fashion and written in standard English?

Reviewer #1: Yes

Reviewer #2: Yes

5. Review Comments to the Author

Reviewer #1: I enjoyed reviewing the Accola et al. paper. It is abundantly clear that the authors collected a lot of data and carefully analyzed it to assess juvenile Chinook salmon diets, energy density of their prey, and metabolic demands in five locations that are important salmon habitat in the Columbia River. The paper is very well written – especially the introduction and discussion. I found no major issues with the paper, but have several comments that will increase the readability and interpretability.

Key words: consider the overlap between key words and title. The authors may want to change some of the key words to have a wider search response

L13: “is” should be past tense “was” to match the rest of the sentence.

L17: “river discharge” should have some more information. Maintaining adequate river discharge? What sort of river discharge. Sample with L18 connectivity. Increased connectivity?

Fig 1 should have a scale bar. It might also be useful for international readers if there were an inset of a broader geographical region. That part is optional but might help people interpret the location of the LCRE

L103 and L107 spell Campbell differently

L117: this would be a good point to mention the IACUC protocols used and collection permits

L118: what level were prey identified to? Genus? Broader taxonomic groupings? I see on L119 it says family, genus, or species. How was that decided? Lowest possible taxonomic unit? Did it vary depending on whether it was a macroinvertebrate or zooplankter? Was the entire stomach contents counted or a fraction of it and then mathematically scaled to the whole thing?

Methods: there is no information about how the water quality data were collected. The first part of the results includes temperatures, but the reader is not sure whether these are surface temperatures, buoy data, or continuous loggers deployed at some reasonable sampling depth.

Results: the water temperature is interesting, but sort of an island. Maybe a figure if they are continuous monitoring data would be prudent? I see it in Figure 6 now that I’ve reached that part, but it is not referenced in text L177-181 where it’s mentioned. And Fig 6 also doesn’t include Fraz Lake and Ilwaco Slough

Table 1: It is very impressive that the authors had such a large sample size!

All figures: I printed the manuscript to read and the colors are not distinguishing very well in black and white. Are the colors selected for the figures color-blind friendly? R has many great options for printer and color-blind friendly palettes. The authors could check out colorbrewer2.org or the viridis color schemes in the ‘viridis’ R package. Fig 5 uses a colorbrewer theme, however the ‘stomach fullness’ ranges for each color are not defined in the legend, which would be helpful for the reader.

For Fig 2, it is hard to interpret the lines. The authors could consider stacked bar graphs here to show the relative importance of different prey items. I can see that Amphipoda and Diptera seem to be the most important and fish and Trichoptera the least, but stacked bar graphs may be more interpretable.

Fig 3 is a neat way to present the data.

I really like Fig 4 too.

For Fig 5, it is a bit unclear whether the y-axis “Juvenile Chinook salmon counts” means the number of stomachs assessed or some sort of population per unit area estimate. The latter would be really interesting to see if stomach fullness was related to competition.

L433-439 seem tangential and should not be the final information a reader gets from this manuscript. The authors should move this to a different part of the discussion and end on their most major and broad-sweeping conclusions, or eliminate it all together.

Reviewer #2: Review of PONE-d-25-06245

This study assesses diets of juvenile Chinook salmon in the Columbia River estuary, USA at five sites over a 14 year period to quantify spatial and temporal trends in prey consumption, stomach fullness, energy consumption, and the metabolic costs associated with fish size and water temperatures. Juvenile Chinook salmon diets were stable, and stomach fullness and caloric intake is comparable among the sites where most salmon were collected. Juvenile Chinook salmon were frequently in water temperatures above fitness thresholds. Higher water temperatures raise metabolic rates, so increased foraging will be necessary for growth in rising water temperature regimes. Reduced growth, earlier migration, and prey production timing mismatches are near term possibilities. The authors speculated that river discharge, habitat restoration and connectivity can aid rearing resiliency in the estuary.

General comments

I thought this paper was a well-written summary of a long-term study investigating juvenile Chinook salmon diets in the heavily managed Columbia River estuary. The statistical analysis, tables, and figures were appropriate.

I have two major concerns with the paper. First, the authors don’t present testable hypotheses embedded in ecological theory to drive the analysis. As it stands, the paper is just a statistical description of the data. Second, there was no attempt to correlate river discharge or climate indices (ENSO) in driving some of temporal variation in diet composition despite having a 14 year data set.

Specific comments

Line 30: I suggest adding more details about the Columbia River here after you first mention the system.

Line 52: I suggest adding detail about the LCEP here.

Line 97: Please define ‘good water quality’.

Page 118-119: How did you deal with unidentifiable items? I’ve analyzed a lot of fish diet samples and there is a lot of body parts, etc that are not easily identifiable without DNA analysis. This material can dominate the stomachs of some fish. What about fish with empty stomachs?

Line 176: Why no data on discharge? Seems like this is an important driver of diets and fish populations but it was not explicitly evaluated despite being mentioned a lot by the authors.

Line 402: You did not assess fish growth so you cannot say your study provides insight into how tidal wetlands influence this process.

6. PLOS authors have the option to publish the peer review history of their article (what does this mean? ). If published, this will include your full peer review and any attached files.

**Do you want your identity to be public for this peer review?** For information about this choice, including consent withdrawal, please see our Privacy Policy .

Reviewer #1: No

Reviewer #2: No

---

## [Author Response · Author response to Decision Letter 1]

5 May 2025

We have responded to each reviewer and editor comment in the attached document, "Response to Reviewers".

---

## [Editor Report · Decision Letter 1]

Subyearling Chinook salmon diets in Lower Columbia River estuarine habitats

PONE-D-25-06245R1

Dear Dr. Accola,

We’re pleased to inform you that your manuscript has been judged scientifically suitable for publication and will be formally accepted for publication once it meets all outstanding technical requirements.

Kind regards,

Jason Magnuson, Ph.D.

Academic Editor

PLOS ONE
---

## [Editor Report · Acceptance letter]

PONE-D-25-06245R1

PLOS ONE

Dear Dr. Accola,

I'm pleased to inform you that your manuscript has been deemed suitable for publication in PLOS ONE. Congratulations! Your manuscript is now being handed over to our production team.

Kind regards,

on behalf of

Dr. Jason Magnuson

Academic Editor

PLOS ONE